# Massively targeted evaluation of therapeutic CRISPR off-targets in cells

Xiaoguang Pan[1,2,12], Kunli Qu[1,2,3,12], Hao Yuan[1,4,12], Xi Xiang[1,3,12], Christian Anthon [5,12], Liubov Pashkova [5], Xue Liang[1,2], Peng Han [1,2], Giulia I. Corsi [5], Fengping Xu[1,4,6], Ping Liu[6,7], Jiayan Zhong[6,7], Yan Zhou[3], Tao Ma[6,7], Hui Jiang[6,7], Junnian Liu[1], Jian Wang[6], Niels Jessen [3,8], Lars Bolund[1,3], Huanming Yang[6,9], Xun Xu [6,10], George M. Church[11 ✉], Jan Gorodkin [5 ✉], Lin Lin [3,8 ✉] & Yonglun Luo [1,3,4,6,8,9 ✉]

Methods for sensitive and high-throughput evaluation of CRISPR RNA-guided nucleases (RGNs) off-targets (OTs) are essential for advancing RGN-based gene therapies. Here we report SURRO-seq for simultaneously evaluating thousands of therapeutic RGN OTs in cells. SURRO-seq captures RGN-induced indels in cells by pooled lentiviral OTs libraries and deep sequencing, an approach comparable and complementary to OTs detection by T7 endonuclease 1, GUIDE-seq, and CIRCLE-seq. Application of SURRO-seq to 8150 OTs from 110 therapeutic RGNs identifies significantly detectable indels in 783 OTs, of which 37 OTs are found in cancer genes and 23 OTs are further validated in five human cell lines by targeted amplicon sequencing. Finally, SURRO-seq reveals that thermodynamically stable wobble base pair (rG•dT) and free binding energy strongly affect RGN specificity. Our study emphasizes the necessity of thoroughly evaluating therapeutic RGN OTs to minimize inevitable off-target effects.

[1] Lars Bolund Institute of Regenerative Medicine, Qingdao-Europe Advanced Institute for Life Sciences, BGI-Qingdao, BGI-Shenzhen, Qingdao, China. [2] Department of Biology, Copenhagen University, Copenhagen, Denmark. [3] Department of Biomedicine, Aarhus University, Aarhus, Denmark. [4] College of Life Sciences, University of Chinese Academy of Sciences, Beijing, China. [5] Center for non-coding RNA in Technology and Health, Department of Veterinary and Animal Sciences, Faculty of Health and Medical Sciences, University of Copenhagen, Frederiksberg, Denmark. [6] BGI-Research, BGI-Shenzhen, Shenzhen, China. [7] MGI, BGI-Shenzhen, Shenzhen, China. [8] Steno Diabetes Center Aarhus, Aarhus University Hospital, Aarhus, Denmark. [9] IBMC-BGI Center, the Cancer Hospital of the University of Chinese Academy of Sciences (Zhejiang Cancer Hospital), Institute of Basic Medicine and Cancer (IBMC), Chinese Academy of Sciences, Hangzhou, Zhejiang 310022, China. [10] Guangdong Provincial Key Laboratory of Genome Read and Write, BGI-Shenzhen, Shenzhen, China. [11] Department of Genetics, Blavatnik Institute, Harvard Medical School, Boston, MA, USA. [12] These authors contributed equally: Xiaoguang Pan, Kunli Qu, Hao Yuan, Xi Xiang, Christian Anthon. ✉email: gchurch@genetics.med.harvard.edu; gorodkin@rth.dk; lin.lin@biomed.au.dk; alun@biomed.au.dk

Clustered Regularly Interspaced Short Palindromic Repeats (CRISPR) RNA-guided nucleases (RGNs) has been used in therapy of several inherited human diseases[1–4]. Major efforts have focused on improving RGN editing efficiency via stabilization of the small guide RNA (sgRNA) thermodynamics[5], modification of the RGNs[6–8], utilization of homology-independent mediated targeted integration (HITI)[9] and optimization of RGN delivery[10,11]. The inevitably adverse effects caused by unspecific RGN editing of cancer genes are major concerns for the clinical application of RGN-based therapies. Improvement of RGN specificity and development of methods for identifying and evaluating the potential off-targets (OT) introduced by RGNs are equally essential to advance RGN-based gene therapy. Several experimental RGN OT identification/quantification methods have been developed (Supplementary Data 1), which can be grouped into three categories (Supplementary Fig. S1). Category One contains genome-wide cell-free biochemical methods which relies on the capture of RGN-induced OT cleavage on either naked DNA or fixed chromatin fibers by sequencing. Examples are CIRCLE-seq (cell-free)[12], Digenome-seq (cell-free)[13], SITE-seq (cell-free)[14], BLISS (ex vivo)[15] and DIG-seq (ex vivo)[16]. Category Two contains methods depending on genome-wide in-cell capturing of RGN-induced off-target cleavage by sequencing, such as GUIDE-seq and IDLV-capture relying on insertion of double strand DNA and IDLV vector to the DNA double strand breaks respectively[17,18], HTGTS and PEM-seq relying on translocation between on-target and off-targets[19,20], and DISCOVER-seq relying on immunoprecipitation of DNA repair protein MRE11 to capture the DNA double strand break (DSB) sites[21]. While cell-free biochemical methods are rapid, conventional, and not depending on reference genomes, they inevitably capture many pseudo off-target sites. In-cell methods (e.g., GUIDE-seq) capture the bona fide RGN off-targets more faithfully as compared to cell-free methods. However, spontaneous DSBs lead to capturing pseudo off-targets independent of RGNs[17]. To complement this, Category Three is composed of targeted in-cell RGN OT validation methods, such as T7 endonuclease 1 (T7E1), targeted deep sequencing, TIDE and CUT-PCR[22,23]. However, current targeted in-cell RGN off-target evaluation methods are greatly limited by their scales. Only a few sites can be evaluated for each RGN in a single study due to their high labor and time cost. A modified targeted amplicon sequencing method based on the rhAmpSeq has thus been reported for simulated and targeted analysis of several CRISPR gRNAs and hundreds of selected off-target sites in a single reaction[24,25]. This method has greatly improved the scale of targeted analysis of CRISPR off-targets by deep sequencing.

Here we introduce and apply SURRO-seq, a high-throughput method for targeted in-cell capture of RGN off-targets based on a pooled lentiviral vectors library encoding gRNA and barcoded surrogate off-target sites, to evaluate therapeutic RGN off-targets in cells. SURRO-seq exhibits high sensitivity and accuracy compared to GUIDE-seq and CIRCLE-seq by evaluating 170 previously investigated OTs from 11 RGNs in HEK293T cells. We then applied SURRO-seq to evaluate 8150 OTs from 110 therapeutic RGNs and identify 783 OTs showing significantly detectable indels. 37 OTs with significantly detectable indels are found in cancer genes, highlighting the clinical significance and great need of pre-assessing RGN OTs with SURRO-seq. The SURRO-seq identified OTs were further validated by targeted deep sequencing of five RGN-edited human cell lines. Analyses of OTs with high indel frequencies revealed that mismatch types leading to thermodynamically stable wobble base pair strongly increase RGN OT effect. We further perform benchmark analyses of latest RGN OT prediction tools with SURRO-seq OT data. The energy-based predictors, which incorporate gRNA and DNA binding energies, give the best performance.

## Results

**Design of SURRO-seq.** Libraries of surrogate vectors have been used in many studies to massively capture on-target efficiencies[26,27]. We and others show that single surrogate episomal vector (or genomic integration site) has been used as a sensitive method to measure RGN off-target activity. But the method is only applicable for evaluating a limited number of RGN OTs[7,28]. Previously, we introduced an optimized high-throughput approach for targeted in-cell evaluation of on-target RGN efficiency using a pool of lentiviral surrogate vectors[29]. Here we introduce site specific barcoding and repurpose the method for high throughput targeted evaluation of RGN off-targets (OTs) in cells. For a given RGN, protospacer sequences of all OTs are very similar and only differ for 1–5 nucleotides (nt). Following RGN editing, deletion indels could erase nucleotides that differ between OTs, making it impossible to uniquely assign the deletion indels to the OTs (Supplementary Fig. S2). To overcome the indel split problem, we introduced a 10-nt barcoding strategy to distinguish indels reads in the ON and OT sites (Supplementary Fig. S2). As showed in Fig. 1, SURRO-seq contains nine major steps (see Supplementary Note 1 for extended description of the method) with three modifications compared to our previous on-target method CRISPRon[29]: (1) The surrogate site contains a 10-nt barcode preceding the 27-nt surrogate OT site, which contains the OT protospacer (20 nt), protospacer adjacent motif (PAM, NGG), 4-nt PAM downstream sequences; (2) Barcode-guided split of indel reads (Supplementary Figs. S2, S3); and (3) Fishers' exact test of OTs with significant indels [a. Two-fold higher indel frequency in the SpCas9 cells as compared to the wild type cells (MOCK); b. Fishers exact test and Benjamini and Hochberg (BH)-adjusted $p$-value less than 0.05] (Supplementary Note 2).

**Validation of previously evaluated RGN OTs with SURRO-seq.** First, we sought to assess if SURRO-seq can capture RGN OTs previously evaluated by other methods. We generated a small library (LibA) containing 170 OTs from 11 RGNs (Fig. 2a). These 11 RGNs and 170 OTs had been detected by T7E1[30,31], GUIDE-seq[17] and/or CIRCLE-seq[12]. We transduced SpCas9-overexpressing (SpCas9) and wildtype (MOCK) HEK293T cells with LibA (MOI = 0.3, and 4000-fold coverage of LibA, see methods). Eight days after LibA transduction, indel frequencies introduced in the surrogate OTs were quantified by targeted deep sequencing. Analyses of LibA data (Supplementary Fig. S3–5, Fig. 2b, Supplementary Data 2) showed that SURRO-seq can capture nearly 100% of the T7E1-detected OTs (22 out of 23, Fig. 2b), most (104 out of 149, 70%) of GUIDE-seq-captured OTs (Fig. 2c), and approximately half (78 out of 153, 51%) CIRCLE-seq-captured OTs (Fig. 2d). Five RGNs have been analyzed by GUIDE-seq, CIRCLE-seq and SURRO-seq. Comparison of these 141 OTs from the five RGNs showed that a large subset (82 OTs, 58%) of these 141 OTs are captured by at least two methods (Fig. 2e). There are 1, 18, and 40 OTs only captured by SURRO-seq, GUIDE-seq and CIRCLE-seq respectively (Fig. 2e). CIRCLE-seq is based on CRISPR-Cas9 cleavage of cell-free and histone-free DNA, and GUIDE-seq is relying on repair of DSBs and insertion of the targeted double-strand DNA oligonucleotide in cells. Since the chromatin can inhibit Cas9 off-target effect[16], it is thus not surprising to observe that a subset of these RGN OTs can only be captured by one method[12,13,17]. SURRO-seq offers a complementary in-cell method for targeted validation of RGN OTs identified by genome-wide screening approaches.

**Large-scale evaluation of therapeutic RGNs OTs with SURRO-seq.** To investigate if SURRO-seq can be used for high throughput targeted in-cell evaluation of RGN OTs, we selected 110 RGNs

## An overview of the SURRO-seq - based CRISPR off-target evaluation

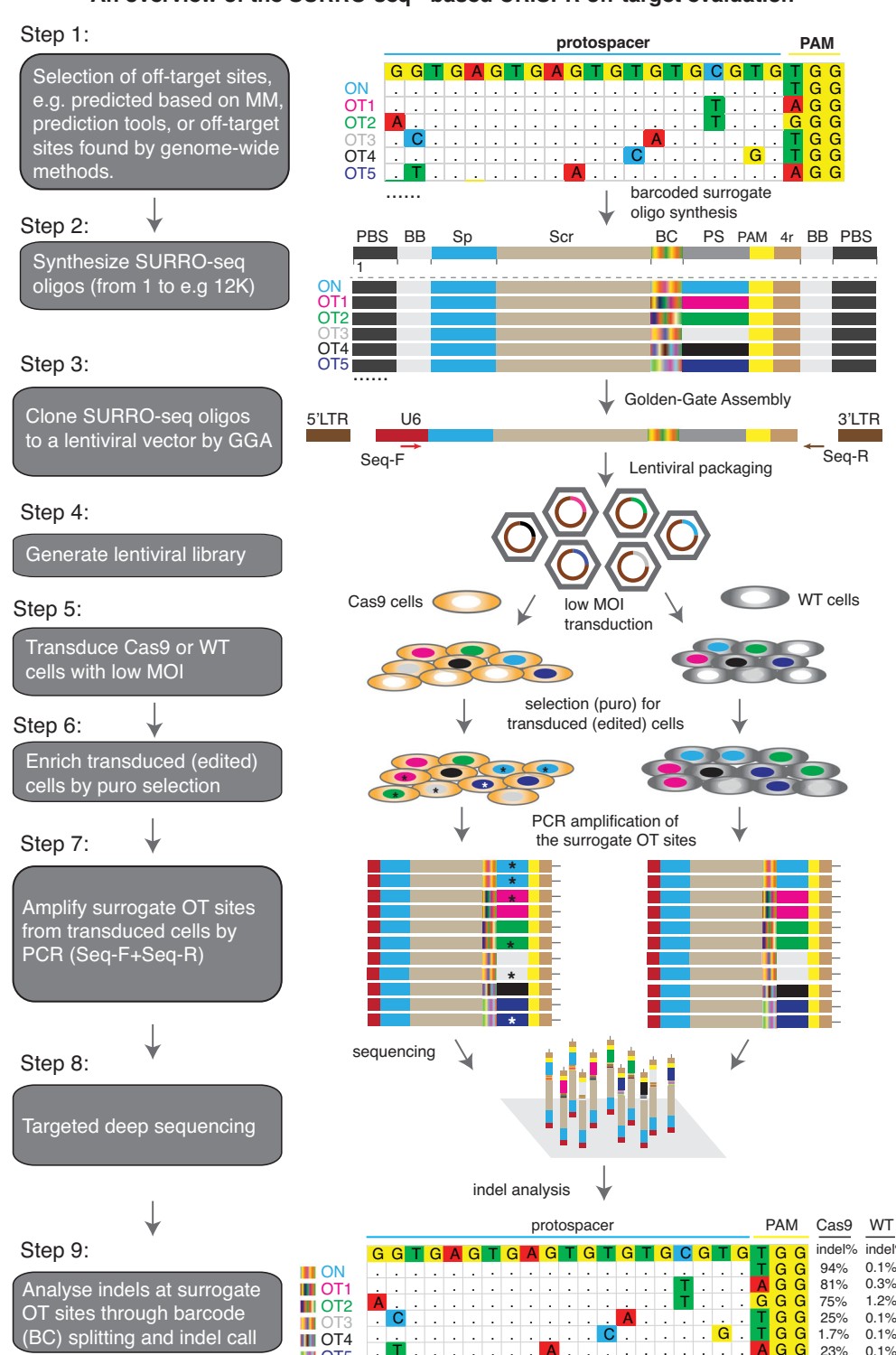

**Fig. 1 Design of SURRO-seq.** An overview of the nine major steps of SURRO-seq is schematically presented. MM, mismatches; ON, on-target; OT, off-targets; PBS, primer binding sites; BB, BsmBI binding site; Sp, spacer; Scr, SpCas9 gRNA scaffold; BC, barcodes; PS, protospacer; PAM, protospacer adjacent motif; 4r, 4 bp downstream sequences; GGA, Golden Gate Assembly; MOI, multiplexity of infection; WT, wildtype; Results presented in Step 9 are indel frequencies captured by SURRO-seq for RGN *VEGFA* T3.

targeting 21 human genes that have been used in preclinical gene therapy studies (Supplementary Data 3). Given that most RGNs can only tolerate a few (1–3) mismatches between the sgRNA spacer and the protospacer sequences[32–34], we blasted the human reference genome with the 20-nt spacer of each RGN and retrieved all potential OTs with up to 4 mismatches. In total, 8150 OTs from these 110 RGNs were selected and synthesized, cloned into the SURRO-seq vector, and packaged into lentivirus,

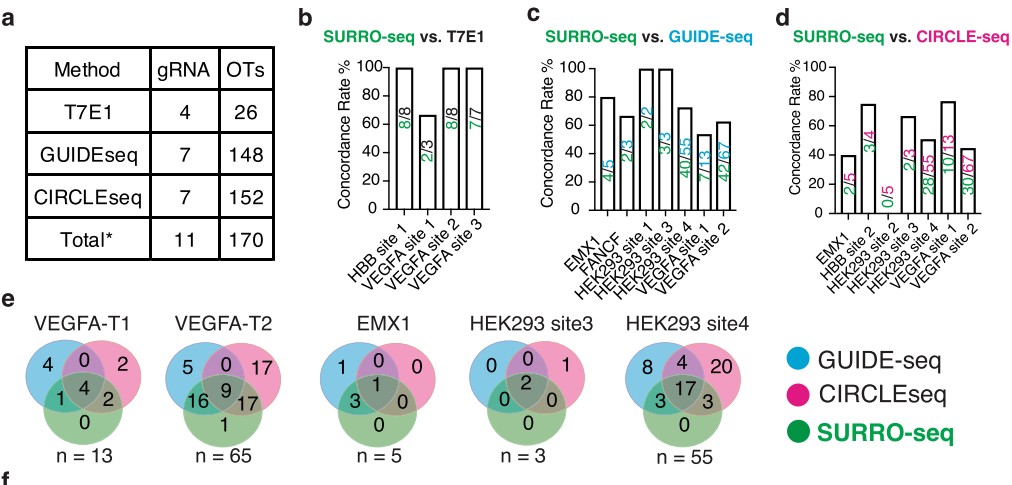

**f  Comparison of VEGFA-T1 OTs detection between T7E1, GUIDEseq, CIRCLEseq and SURRO-seq**

**g  Comparison of VEGFA-T2 OTs detection between T7E1, GUIDEseq, CIRCLEseq and SURRO-seq**

(Total VEGFA-T2 OTs data were shown in Extended Tables S2)

hereafter referred to as library B (LibB) (Fig. 3a). We transduced SpCas9 and wildtype (MOCK) HEK293T cells with LibB (MOI = 0.3, 4000-fold coverage) and analyzed indel frequencies from cells eight days after transduction by deep sequencing.

We first analyzed the on-target gRNA efficiency of these 110 RGNs. SURRO-seq successfully captures on-target efficiencies for all 110 RGNs (Fig. 3b). The SpCas9 protein is overexpressed in the HEK293T cells by doxycycline addition. Consistent with our previous observation[29], most (n = 96) RGNs exhibited high on-target activity (indel frequencies% (IF%) > 80%, Fig. 3b). A few RGNs (n = 14) had relatively low efficiency (IF% < 80%), and these were also significantly (p < 0.0001) lower in GC content compared to highly efficient RGNs (IF% > 80%) (Supplementary Fig. S6). Next, we analyzed indel frequencies in the OTs

**Fig. 2 Validation of RGN OTs detection between T7E1, GUIDE-seq, and/or CIRCLE-seq by SURRO-seq. a** Overview of RGN gRNAs and OTs selected for validated with SURRO-seq. **b**–**d** Comparison of the OT detection concordance rate between SURRO-seq and T7E1 (**b**), GUIDE-seq (**c**) and CIRCLE-seq (**d**). Numbers are total OTs for each RGN (upper) evaluated with the compared method and OTs agreed with SURRO-seq (lower). **e** Venn diagram comparison of OTs with significant significantly detectable off-targets (SURRO-seq) and OTs with deep sequencing reads detected by GUIDE-seq or CIRCLE-seq. Numbers are OT sites. **f**–**g** Comparison of VEGFA-T1 (**f**) and VEGFA-T2 (**g**) OT detections between T7E1, GUIDE-seq, CIRCLE-seq and SURRO-seq. Full results are showed in Supplementary Data 2. RGI, relative gel intensity; RE%, percentage of relative efficiency, calculated by % reads in OT per reads in ON; IF, indel frequency; I/T reads, indel/total reads; $P$ values for comparison between SpCas9 and MOCK IF% are calculated with Benjamini and Hochberg (BH)-adjusted Fisher's exact test (two-sided). *, represents OT with significantly detectable indels (adj. $P$ value < 0.05, FC (IF% SpCas9/ IF% MOCK) > = 2). NS, represents OTs with not significantly detectable indels.

introduced by RGNs. Surrogate OT sites with low sequencing quality (total clean reads <32 for both MOCK and SpCas9), low synthetic quality (IF% in MOCK > 4%, Supplementary Fig. S7) were filtered. Mutagenesis in essential genes caused by random integration of the SURRO-seq lentiviral vector or by OT targeting could affect cell proliferation. We performed surrogate site enrichment and enrichment analysis (Supplementary Fig. S8) and identified 196 sites (Fold-change between MOCK and SpCas9 > 2). Significantly detectable indels (fold-change (FC) of IF% SpCas9 / IF% MOCK > 2, adj. $p$-value < 0.05) were found in 30 of these RGN OTs and strikingly all depleted (Supplementary Fig. S8C, Supplementary Data 3). Nine RGN OTs (Supplementary Data 3) are known human essential genes by mapping to the human essential gene database DEG10[35]. As these depleted/ enriched surrogate sites might affect the later analysis of the effect of DNA context and thermodynamics on RGN OT activity, we also excluded these sites for subsequent analysis. After the filtering, 7140 OTs were retained for downstream analyses (Supplementary Data 3). Quantification of indel frequencies in each OT in SpCas9 and MOCK cells showed that there were not significantly detectable indels for most of these OTs ($n = 6387$, hereafter referred to as NSOT). Significantly detectable indels (fold-change (FC of IF% SpCas9/IF% MOCK) > 2, adj. $p$-value < 0.05) were identified for 753 OTs in SpCas9 cells compared to MOCK (Fig. 3c, Supplementary Data 3). However, the indel frequency of most Sig. OTs were less than 3% (573 out of 753, Fig. 3c, d). We further divided the Sig. OTs into two groups: based on IF% in SpCas9 < 3% (Low Indel Sig. OTs, hereafter referred to as LIOT) and IF% > = 3% (High Indel Sig. OTs, hereafter referred to as HIOT). Notably, most HIOTs contain 1–3 mismatches (Supplementary Fig. S9). Our results demonstrate that the SURRO-seq can be used for high throughput targeted evaluation of RGN-induced indels at surrogate off target sites in cells.

To investigate where were these LIOTs and HIOTs located in the genome and in genes, we annotated their genomic locations according to the presence in intergenic region (IGR) or in genes (2 kb upstream, 5' untranslated region, exon, intron, 3' untranslated region, 2 kb downstream; Supplementary Data 3). Despite that most of the Sig. OTs are found in intron and IGRs, there are still a substantial number of Sig. OTs (nr. of LIOT = 200, nr. of HIOT = 87) found in gene exons and/or regulatory regions that might affect gene expression (Fig. 3d, Supplementary Fig. S10). Notably, 37 Sig. OTs were annotated in cancer-related genes (Supplementary Data 3). The RGN11153 (spacer sequences, CTGCTGCTGCTGCTGCTGGA), which was proposed for Huntington's Disease therapy by targeting the CAG expansion tract[36], exhibit great off-target effect (nr. of LIOT = 43, nr. of HIOT = 35). Two HIOTs of RGN11153 are found in cancer genes *ZFHX3* (exon) and *SOHLH2* (intron). The zinc-finger homeobox 3 (*ZFHX3*) is a tumor suppressor gene and knockout of *ZFHX3* in mouse leads to development of neoplastic lesions. Loss of function mutations in *ZFHX3* are frequently detected in human cancers i.e. high-grade human prostate cancers[37],

endometrial cancers[38], urothelial bladder carcinoma[39], lung and brain tumors[40]. This finding emphasizes that carefully evaluating if therapeutic RGNs introduced any off target indels in cancer genes is needed. HIOTs were also found in another two cancer genes *BCOR* (intron) and *NCOR2* (intron) by RGN11155 and RGN11189 respectively. These two RGNs were used for HD (RGN11155) and β-Thalassemia (RGN11189) therapy[41]. For the LIOTs, despite low indel frequency (below 3%), significantly detectable indels were found in the exon of nine cancer genes, such as *U2AF2* and *NKTR* causing Acute myeloid leukemia (AML) (Supplementary Data 3 and Fig. S10). SURRO-seq thus offers a high throughput and targeted approach for in cell evaluation of RGN OTs in cells.

**Validation of SURRO-seq identified OTs by targeted deep sequencing of endogenous genomic loci.** To validate if OTs captured by SURRO-seq were also presented at the corresponding endogenous sites, we analyzed 23 OTs from seven RGNs in five human cell lines: human embryonic kidney cells (HEK293T), human primary fibroblasts, lung cancer cells (PC-9), ovarian cancer cells (SKOV3), and bone osteosarcoma epithelial cells (U2OS). Of these 23 OTs, 16 and 7 OTs were detected with significant and non-significant indels by SURRO-seq, respectively (Fig. 4a and Supplementary Data 4). These 16 Sig. OTs were selected to cover a broader distribution of indel frequencies, ranging from 2% to 96%. Instead of using lentivirus-based delivery of CRISPRs, we applied an optimized CRISPR delivery approach based on CRISPR ribonucleoprotein (RNP). Highly efficient delivery of CRISPR into various types of cells have been reports by us and many other groups[2,5,42,43]. We also validated that an enhanced green fluorescent protein (*EGFP*) mRNA can be delivered to nearly 100% of cells in all five cell lines (Supplementary Fig. S11). Seven on-target sites and 23 off-target endogenous genomic loci from the five cell lines were analyzed by targeted deep sequencing 48 h after RNP treatments (Fig. 4b). Several Cas9 mutants have been reported with improved specificity[44–46]. In addition to the wild type SpCas9, we also analyzed the indel frequencies in the 7 RGN on-target sites and 23 off-target sites in cells transfected with a high-fidelity Cas9 variant (HiFi-Cas9), of which a single point mutation (p.R691A) was introduced[44].

Analyses of deep sequencing results showed that significant indel frequencies (one-way ANOVA, $p < 0.05$) were detected in all seven RGN on-target sites in the five cell lines (Fig. 4c, Supplementary Fig. S12, Supplementary Data 4). Consistent with early results, HiFi-Cas9 retained similarly high on-target activity as the wild type Cas9 (Fig. 4c), except RGN11208 which is low in GC content (GC% = 20%, Supplementary Fig. S13). Analyses of indel frequencies in the 23 off-target sites showed that there is a good agreement (20 out of 23, 87%) between SURRO-seq and targeted sequencing of endogenous sites. 15 out of 17 (88%) of the SURRO-seq off-target sites with significantly detectable indels were validated by targeted deep sequencing of in Cas9 RNP

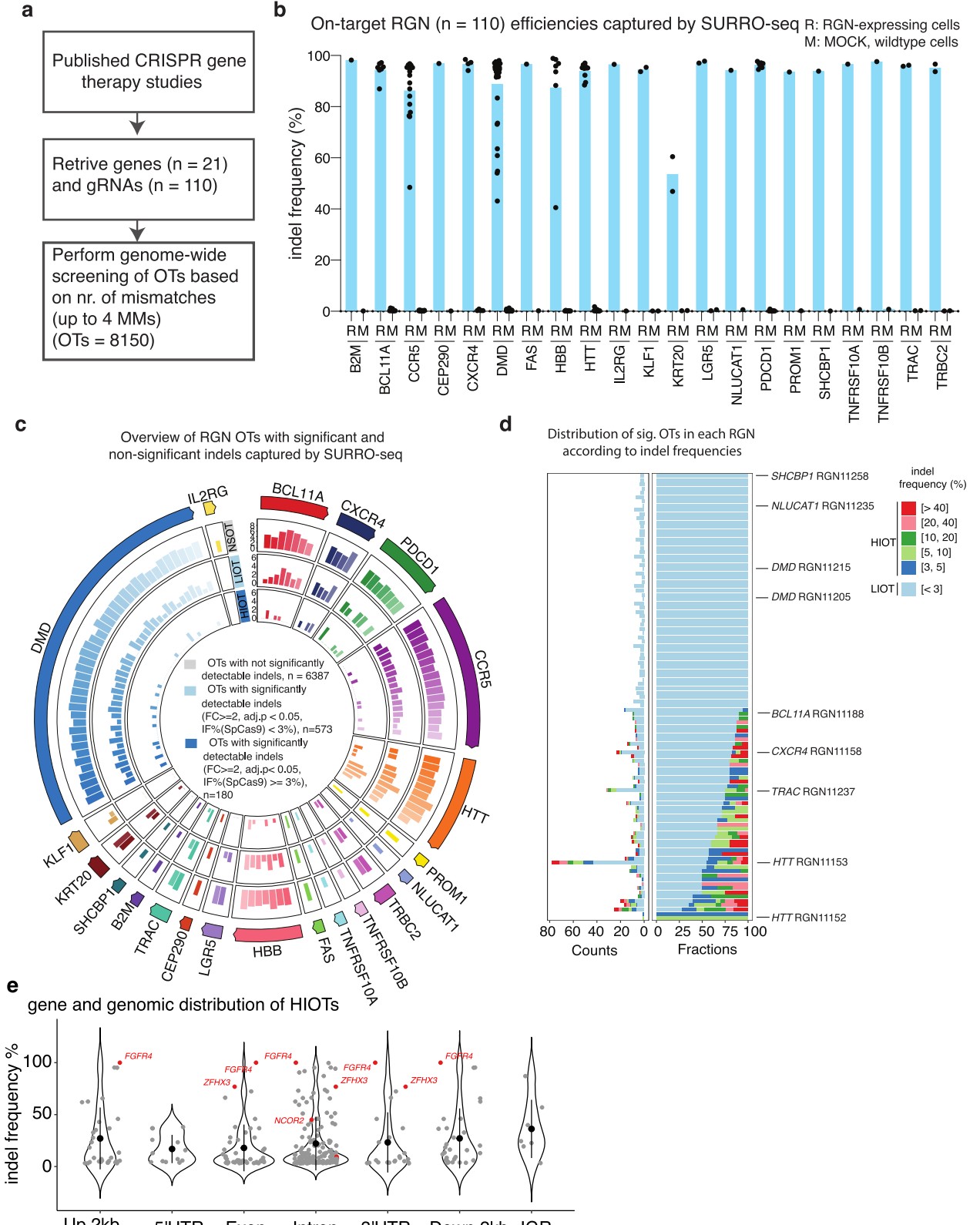

**Fig. 3 High throughput evaluation of gene therapy RGN OTs with SURRO-seq. a** Overview of gene therapy RGN selection and number of OTs captured. **b** Quantification of indel frequencies for the 110 RGNs by SURRO-seq. R, RGN edited, M, MOCK control. **c** Overview of the number of RGNs OTs with not significantly detectable indel (NSOT, outer circle, adj. *P* value > 0.05 or FC (IF% SpCas9/ID MOCK < 2)), with significantly detectable indels (adj. *P* value < 0.05 and FC (IF% SpCas9/ID MOCK >= 2)) but low indel frequency (<3%, LIOT, middle circle), and significantly detectable indels with high indel frequency (>= 3%, HIOT, inner circle). *P* values are derived from Benjamini and Hochberg (BH)-adjusted Fisher's exact test (two-sided). **d** Bar plot of total number (left) and fraction (right) of LIOTs and HIOTs for the RGNs. **e** Violin plot of the gene and genomics location of the HIOTs and indel frequency. OTs in cancer genes are highlighted in red. Data are presented as mean values +/− SD.

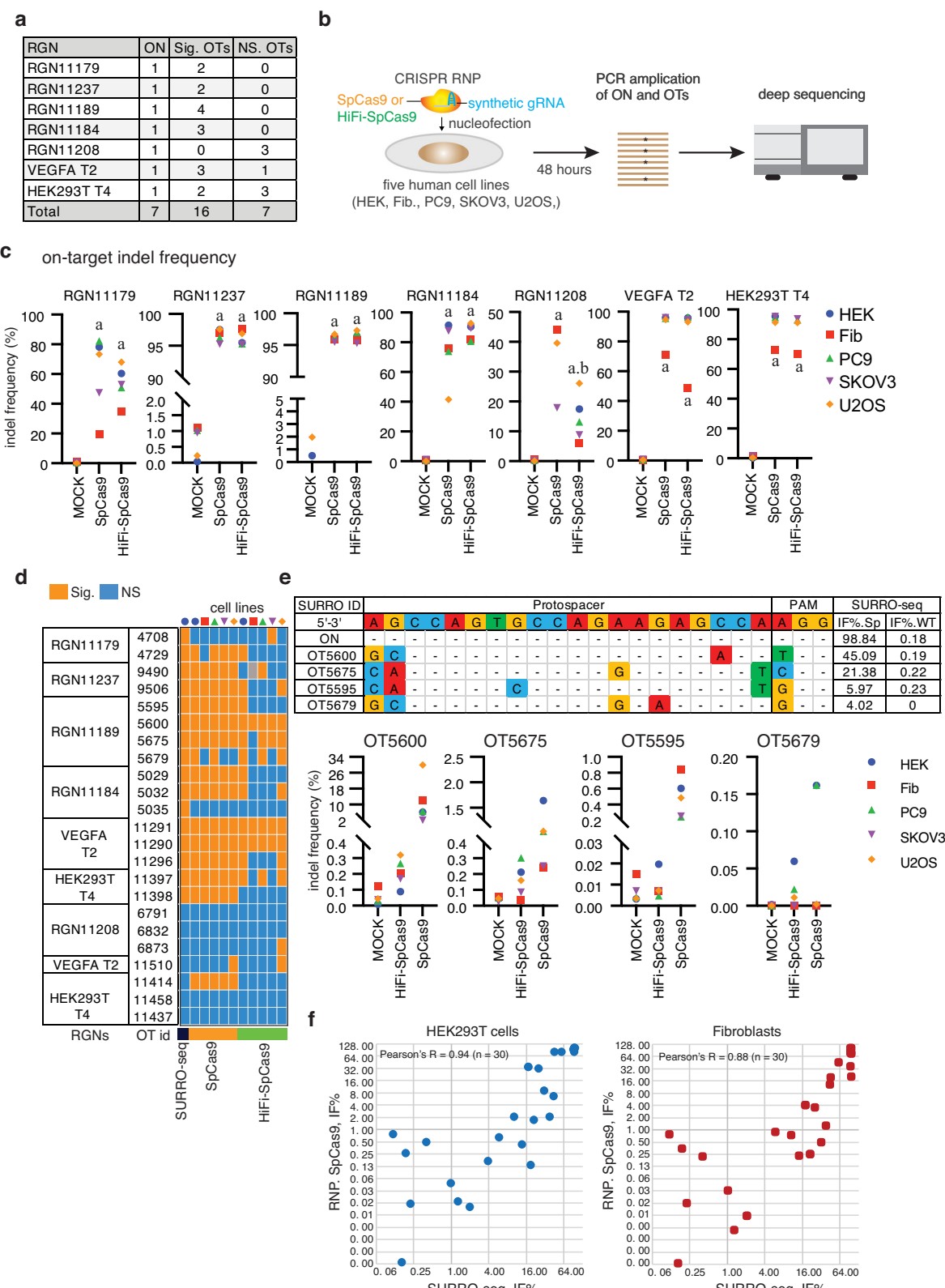

treated cells (Fig. 4d, e, Supplementary Data 4). Due to differences in RGN delivery, editing time and RGN expression level between SURRO-seq and RNP nucleofection, the indel frequencies from the endogenous off-target sites are much lower than that measured by SURRO-seq (Fig. 4e, Supplementary Data 4). Despite that, there is a generally good correlation between the SURRO-seq and the endogenous editing results (Pearson's

R = 0.88–0.94, Fig. 4f, Supplementary Data 4). Most importantly, both the number of OTs with significantly detectable indels and the indel frequency were significantly reduced in cells edited with the high-fidelity HiFi-Cas9, which corroborates with previous finding and highlights the importance and necessity of using high-fidelity Cas9 variants in gene therapy application to minimize the off-target effect[44]. Collectively, we demonstrated

**Fig. 4 Validation of endogenous OTs in five human cell lines by deep sequencing. a** Overview of RGNs, SURRO-seq identified Sig. OTs and NS. OTs selected for validation. **b** Schematic illustration of the experiments. RNP, ribonucleoprotein; HEK, HEK293T cell; Fib, human skin-derived fibroblasts. **c** Dot plot of on-target indel frequencies (indel reads/total reads %) in the CRISPR RNP edited cells. Indel frequency values were showed in Supplementary Data 4. **d** Heatmap summary of the RGN OTs evaluated by SURRO-seq and deep sequencing in five human cells lines. Indel frequency values were showed in Supplementary Data 4. **e** Example of indel frequencies of four OTs for RGN11189 measured by SURRO-seq and by deep sequencing of RGN edited human cell lines. **f** Scatter plots of indel frequencies for 7 on-target and 23 off-targets (referred to 4a), measured by SURRP-seq and by amplicon sequencing of the corresponding endogenous loci in RNP nucleofected cells (HEK293T and Fibroblasts). Extended plots for all cells can be found in Supplementary Data 4 (sheet 4.9).

---

that SURRO-seq is a sensitive method for high throughput targeted evaluation of RGN off-targets in cells.

**Effect of mismatch positions, mismatch types, and free binding energy on RGN specificity.** The RGN off-target data generated by SURRO-seq also allow us to explore how the genomic context affects RGN off-target cleavage. Analysis of indel frequencies of the 753 Sig. OTs (both LIOT and HIOT) showed that indel frequencies were significantly decreased in OTs with more mismatches (Fig. 5a, Supplementary Fig. S10), which corroborates with previous findings and is expected[32,47,48]. While it is generally believed that OTs with 3–4 mismatches are unlikely to be cleaved by RGNs, our results showed that there exists a great heterogeneity in mismatch tolerance among the 110 RGNs and between the OTs with same number of mismatches (Fig. 5a). This phenomenon was also observed for the VEGFA-T2 RGN[17] and validated by SURRO-seq (Fig. 2). We speculated that the heterogeneity of indel frequencies between OTs with same number of mismatches in our dataset is caused by the positions and types of mismatches between the RGN spacer and the target site. It has been well characterized that the CRISPR is less tolerated to mismatches in the PAM-proximal 10–12 nucleotides[49–51].

To address if this position-dependent mismatch tolerance contributes to the heterogeneity of OTs, we analyzed the frequency of mismatches occurred in each position of the 20-nt protospacer region for all RGN OTs with 3 or 4 mismatches (Supplementary Data 5). There is a significant (Hypermetric test $P$ value < 0.05) over-representation of mismatches occurred at N1 and N2 positions (the two most PAM-distal nucleotides) and an under-representation of mismatches occurred at the N12-N18 (PAM-proximal seed regions) in OTs with significantly detectable. Interestingly, our analysis also revealed that RGNs seem to be more tolerant to mismatches at the N19 and N20 position as compared to other nucleotides of the seed region (Fig. 5b).

We next analyzed the effect of mismatch types on RGN OTs. Twelve types of mismatches can occur between RGN and off-target sites (Supplementary Fig. S13). To provide a simple description, we only refer to mismatches between the gRNA spacer and the protospacer sequences (the non-targeting strand). Cumulative studies have suggested that RGN exhibits different tolerances to the different types of mismatches. Once such example is the GA mismatch, which generates a wobble base pair (rG:dT) between gRNA spacer and the complementary strand DNA. We analyzed the frequencies of these 12 mismatch types in the OTs from LibB (Supplementary Data 5). Our results showed that GA mismatch (also AG mismatch but to a lesser degree, for OTs with 3MM) was significantly (hypergeometric test $p$-value < 0.05, compared to NS OTs or total OTs) enriched in the OTs with significantly detectable indel (Fig. 5c). To further validate the effects of mismatch and mismatch type on CRISPR specificity, we generated a small SURRO-seq library (libC) carrying artificially generated OTs with all possible combinations of one mismatch for five RNGs (Supplementary Fig. S14). SURRO-seq-based libC further showed similar findings about the effect of mismatch position and type (GA and AG wobble pairs)

on RGN specificity (Fig. 5d–g, Supplementary Fig. S14, Supplementary Data 5). Notably, the RGN 11157, which is low in GC and particularly G content, seems to be less tolerant of mismatches (Supplementary Fig. S14).

Since a large number of regression-based, machine-learning and deep-learning models have already been developed for in silico prediction of RGN off-targets, we benchmarked six RGN OT scoring models: MIT[52], deepCRISPR[53], Cutting Frequency Determination (CFD) score[47], CROP-IT[54], CCtop[55], and CRISPRoff[56] with the LibB Sig. OTs data. Our results showed that CRISPRoff (Pearson R = 0.50, Spearman R = 0.48, p-value < 0.001) outperforms the other four RGN OT scorers (Supplementary Fig. S15). Compared to the other OT prediction scorers, the CRISPRoff has included the free energy feature. We hypothesized that the energy features are the main contributing factors to the RGN OT prediction. To prove that we next analyzed the correlation between the OT indel efficiencies and position-weighted binding energy between gRNA and the (off-) target DNA ($\Delta G_H$), the free energy of the DNA duplex ($\Delta G_O$), or the folding energy of the gRNA only ($\Delta G_U$) as defined by the CRISPRoff energy model[56]. Our results showed that there is significant correlation between indel efficiencies of Sig. OTs and the $\Delta G_H$ (Pearson's R = 0.53, Spearman's R = 0.52, p-value < 0.001), $\Delta G_O$ (Pearson's R = 0.25, Spearman's R = 0.26, p-value < 0.001), and $\Delta G_U$ (Pearson's R = 0.23, Spearman R's = 0.30, p-value < 0.001) (Supplementary Fig. S16). Notably, the feature of gRNA and the (off-)target DNA binding energy ($\Delta G_H$) yields even high correction compared to the seven RGN off-target predicting, corroborating that $\Delta G_H$ is the major energy feature determining RGN OT effect[56]. Our data collectively highlighted the importance of mismatch positions (where), mismatch types (which), free binding energy (a combined feature of mismatch positions and types) on RGN off-target effect.

## Discussion

In conclusion, we validated and demonstrated that surrogate off-target site-based capturing of RGN cleavage can be used for massively targeted evaluation of SpCas9-based RGN off-targets in cells. Similar to our approach, Fu et al., very recently reported a similar library-based approach of which a pair of on-target and off-target surrogate site was introduced to allow direct comparison of on and off target efficiencies, as well as understanding effect of sequence contexts on RGN specificity[57]. Several generations of CRISPR-derived technologies have successfully reported for gene editing purposes. These include the different classes and types of CRISPR Cas systems and variants, such as SpCas9, SaCas9, NmCas9, Cas-X, Cas-Y, Cas12a, Cas13 (just to mention a few)[51,58–62]. Most importantly, many CRISPR-Cas9 derived genetic and epigenetic editing tools have been developed by fusing the dead Cas9 (dCas9) protein or nickase Cas9 (nCas9) protein to effector proteins or protein domains. By fusing dCas9 or nCas9 to deaminases, the CRISPR-Cas9 system have been repurposed for targeted base editing[62–65], such as A-to-G substitution (ABE), C-to-T substitution (CBE), C-to-G substitution (GBE). For a comprehensive overview of the CRISPR-derived

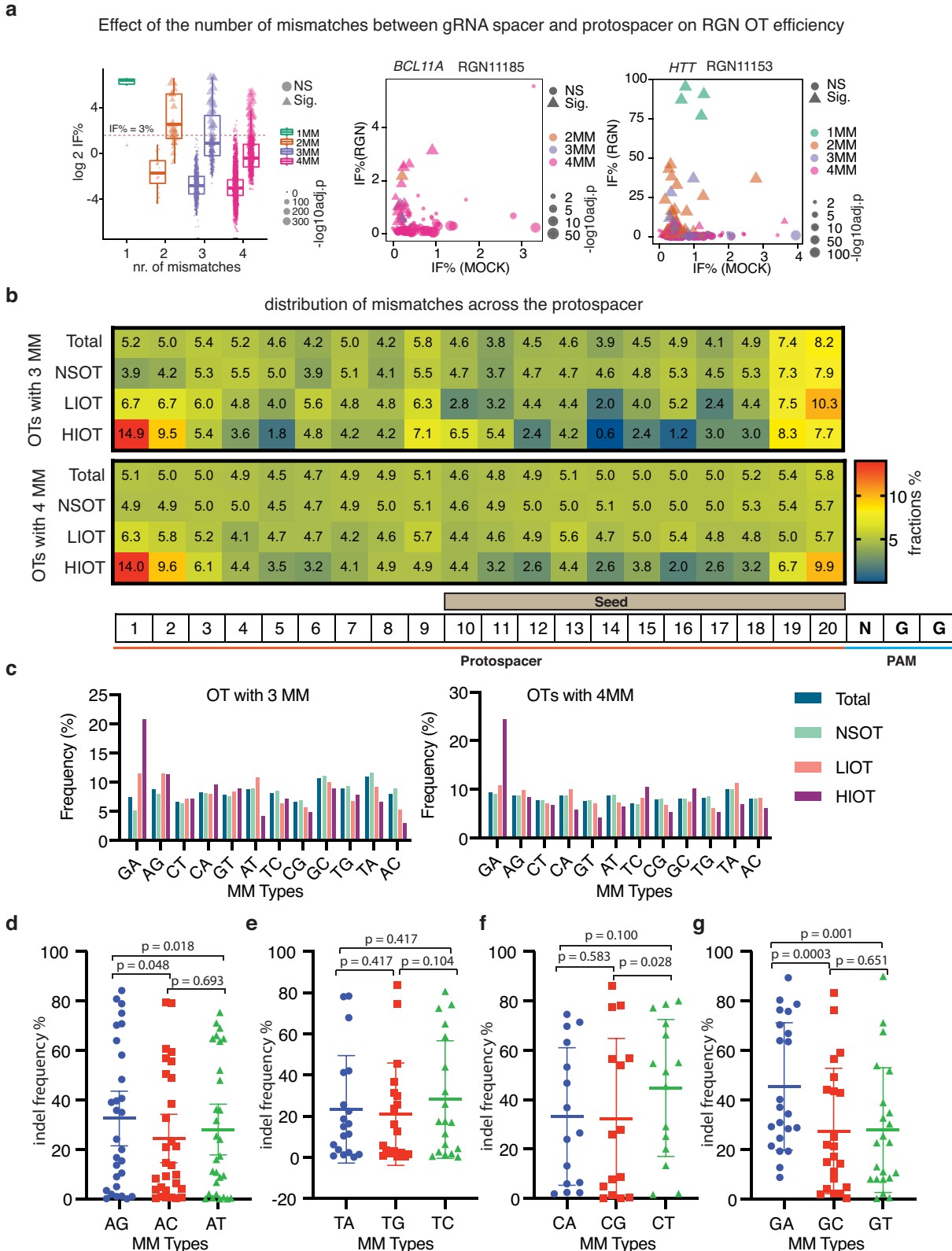

base editors, we refer readers to the review paper by Porto et al.[66]. Off-targets effects have been reported for these CRISPR base editors. Although not showed in this study, we have demonstrated that high throughput quantification of base editing efficiency can also be achieved using such a surrogate library in cells (BioRxiv. https://doi.org/10.1101/2020.05.20.103614). Recently, we further demonstrated that this surrogate library approach can be used to evaluate PAM compatibility in human cells[67]. We anticipate that SURRO-seq could be adapted to evaluate off-targets of other DNA editing RGN systems, including prime editing[68]. Unlike genome-wide in-cell or cell-free OT screening methods, SURRO-seq is limited by its pre-selected potential OTs for evaluation. In this study, we select potential off-target sites for the therapeutic RGNs based on the number of mismatches

**Fig. 5 Effects of mismatch number, position and type on RGN off-target activity. a** Box-and-whisker plot of log 2 indel frequency for RGN OTs evaluated by SURRO-seq in LibB. Data are presented as values representing the median (line within the box), the interquartile range (length of the box), the 75 and the 25th percentiles (whiskers above and below the box) of the indel frequencies. Sites were grouped based the number of mismatches (MM), plotted according to significance and log10 adj. *p*-values (Benjamini and Hochberg (BH)-adjusted Fisher's exact test (two-sided)). NS, RGN OTs with not significantly detectable indels; Sig. RGN OTs with significantly detectable indels. One mismatch (NS, $N = 0$ biologically independent RGN OTs; Sig, $N = 6$ biologically independent RGN OTs), two mismatches (NS, $N = 26$ biologically independent RGN OTs; Sig, $N = 49$ biologically independent RGN OTs), three mismatches (NS, $N = 501$ biologically independent RGN OTs; Sig, $N = 140$ biologically independent RGN OTs), and four mismatches (NS, $N = 5860$ biologically independent RGN OTs; Sig, $N = 558$ biologically independent RGN OTs). **b** Heatmap presentation of the fraction of mismatches occurred in each position of the gRNA for OTs in LibB, grouped based on total OTs, NSOTs, LIOTs and HIOTs. **c** Bar plot of appearance frequencies of each type of mismatches occurred in the different groups of RGN OTs in LibB. **d–g** Dot plots of indel frequencies for OTs with one mismatch measured in LibC. One-way pair-wise ANOVA analysis was performed for A type mismatches (**d**, $N = 30$ biologically independent mismatch sites), T type mismatches (**e**, $N = 20$ biologically independent mismatch sites), C type mismatches (**f**, $N = 18$ biologically independent mismatch sites), and G type mismatches (**g**, $N = 28$ biologically independent mismatch sites). Data are presented as mean values $+/-$ SD. Indel frequency values can be found in Supplementary Data 5 (5.4–5.7).

(allowing up to 4 mismatches). Previous findings from e.g., GUIDE-seq, CIRCLE-seq, as well as validated by SURRO-seq (Fig. 2g) in this study reveal that significantly detectable indels caused by CRISPR-Cas9 can be found in some off-targets with 5 or 6 mismatches. Furthermore, CRISPR off-targets have also been found in genomic sites containing insertions (DNA bulge) or deletions (RNA bulge) compared to the RGN guide sequences[69]. With the development and improvement of in silico RGN off-target prediction tools, just to mention a few e.g., CRISTA[70], CRISPRoff[56], deepCRISPR[53], CNN_std[71] and Elevation[72], it is advisable that the selection of potential RGN OTs for SURRO-seq evaluation should be predicted and selected with these tools. Conversely, more RGN OT data generated with e.g., SURRO-seq or other comparable methods will facilitate the further improvement of these RGN off-target prediction tools. SURRO-seq offers a sound complementary approach to the genome-wide OT screening methods for further high throughput validation of the RGN OTs.

The CRISPR-Cas9 gene editing technology has been in development for a full decade. We still do not completely understand factors affecting its specificity. These specificity-affecting factors include the gRNA-independent binding of the Cas9 protein to DNA, the number/type/position of mismatches between gRNA spacer and the target site, the epigenetic state (DNA methylation and chromatin accessibility), the expression level and duration of the Cas9 protein and gRNA in cells, and the usage of alternative PAMs. Our results suggest that there is a great heterogeneity in term of the specificity among different RGN gRNAs. Corroborating with previous findings[32,33], CRISPR-Cas9 is less tolerant to mismatches at the seed region (N10-N20). Our data further showed that mismatches at the two upstream PAM proximal position (N19 and N20) were more tolerated than other nucleotides of the seed region. This site-dependent effect could be explained by our recent binding energy model about the effect of sliding PAMs on CRISPR-Cas9 specificity[67]. Indeed, when performing benchmarking of the different CRISPR-Cas9 off-target prediction tools with our data, our results also showed that energy-based predictors out-performed other tools in their accuracy of predicting true off-targets. The energy feature is also in agreement with our finding that Wobble base pair (G-U), which still can provide strong binding between the gRNA and target DNA strand, is tolerated. We therefore recommend the use of energy-based tools for in silico prediction of CRISPR potential off-targets, while future further improvements of their prediction outcome should be achieved with high quality off-target data and the integration of better energy features.

Substantial off target indels were observed for some OTs evaluated in this study when conducted in cell line expressing high level of the wild type SpCas9 protein. However, the level of

indels were significantly reduced when the SpCas9 was transiently expressed in cells by RNP delivery. Most importantly, with high fidelity SpCas9 variant (HiFi-SpCas9)[44], our results showed that near all off-target indels could not be significantly detected. Thus, our results strongly indicate that high fidelity SpCas9 variants should be used to its largest extend to avoid any potential adverse effect caused by off-target cleavage. This is particularly important when the CRISPR-Cas9 technology is used for gene therapy, both ex vivo and in vivo deliveries. One remaining major concern of CRISPR gene therapy is the off-target effect leading to oncogenesis due to off-target in cancer genes. Selection of high-fidelity Cas9 variants, carefully design of gRNA with less likelihood of introducing off-target indels in cancer genes, and experimentally validate these potential off-target sites RGN-edited cells are important for lowering the risk of detrimental off-targets in clinical application of RGN. While RGN off-target screening methods, such as GUIDE-seq, DISCOVER-seq, SITE-seq and CIRCLE-seq (also see Supplementary Fig. S1) can be used for genome-wide unbiased detection of RGN off-targets, SURRO-seq overcome the unmet need of high throughput and targeted evaluation of RGN OTs in cells. Our method provides the following four methodological advantages: (1) Scalable. The SURRO-seq library can be generated from a few hundred OTs to over 10,000 OTs. Unlike other methods, SURRO-seq can be used to evaluate hundreds of RGNs in cells simultaneously. (2) Direct evaluation of indels. SURRO-seq directly quantifies the RGN introduced indels at the surrogate off-targets by comparing RGN edited and MOCK cells. (3) High sensitivity. For SURRO-seq, each OT site can be sequenced with a very deep coverage. And direct comparison of indels in RGN and MOCK cells further allow us to sensitively detect OTs with significant indels, and particularly OTs with low indel rate. (4) Clinical significance. SURRO-seq allows us to target evaluate if RGN introduces indels in clinically relevant genes such as cancer genes. However, we also highlight some limitations of SURRO-seq which require further improvement. Each synthetic SURRO-seq oligonucleotide is 170 nt. Synthetic errors introduced in the DNA oligonucleotide library could cause dropout of some OTs after data filtering. Technological improvements in DNA synthesis will overcome this limitation. Alternatively, a two-step cloning strategy can be applied to overcome the length and error-rate limitations of synthetic oligonucleotide pool (Supplementary Fig. S17). First, smaller synthetic oligonucleotides are generated, which contain (a) Two PCR primer binding sites; (b) Two *BsaI* sites (for step1 cloning); (c) RNA spacer; (d) Two *BsmBI* sites (for step 2 cloning) and e. the corresponding surrogate off-target sites. The PCR-amplified surrogate DNAs are cloned to the lentiviral backbone plasmid LentiU6-LacZ-GFP-Puro (BB) (Addgene #170459). Second, the gRNA scaffold stuffer fragment is cloned into the plasmids

generated from Step 1 by Golden-Gate Assembly (*BsmBI*). Another limitation of the technology is caused by the high (average of 1–2%) PCR and/or deep sequencing-induced indel rates observed in wild-type cells. This has limited the detection of potential RGN OTs with low indel frequency. This limitation could be overcome by using improved high-fidelity PCR polymerases and high-fidelity deep sequencing.

As demonstrated in this study, targeted evaluation of RGN OTs in cells with SURRO-seq enables us to investigate the DNA sequences, types of mismatches, and the thermodynamic features on RGN off-target activity. Early studies had shown that other features outside the gRNA (on-target or off-target) sequences such as epigenetic features (e.g., chromatin accessibility, DNA methylation) and gene activity could affect RGN activity and specificity[73–75]. This might partially explain the variations of on-target and off-target activities for those sites observed in the five different cell types (Fig. 4). The surrogate OTs in the SURRO-seq library are randomly inserted in the genome and might not fully capture the epigenetic effects on on-target and off-target RGN activities. Other factors, such as the different presence and preference of DNA double-strand break repair machineries between these cell types (reviewed by Meyenberg M. et al.[76]), could also contributed to RGN activity variations, which however should be addressed in future studies. Although not investigated in this study, we expect that the SURRO-seq method could be used to investigate the effect of e.g., epigenetic factors and DNA repair enzymes on RGN activities (Supplementary Fig. S18). For instance, the SURRO-seq lentiviral library can be stably integrated into wild type (WT) cells. Then, these SURRO-seq WT cells are subjected treatments such as epigenetic modifying molecules (e.g., 5-AZA for DNA demethylation, TSA for histone acetylation), depletion of epigenetic modifying enzymes (e.g., DNA methyltransferases DNMT3A/DNMT3B, histone acetyl transferases P300/CBP), depletion of DNA repair proteins (e.g., DNA ligase 4, XRCC4, MRE11). Thus, SURRO-seq provides an attractive tool for studying factors affecting on-target[29] and off-target RGN activities.

In conclusion, we report a high throughput method for targeted evaluation of CRISPR-Cas9 off-targets in cells. The SURRO-seq offers a great complementary method to the existing tools for CRISPR-Cas9 off-target evaluation, off-target data generation, improvement of prediction, understanding of off-target effect, and facilitate the applications of CRISPR-based gene editing tools in clinical applications.

## Methods

**Cell culture**. Human embryonic kidney (HEK293T), primary human skin-derived fibroblasts (Fib), U2OS, SKOV-3, and PC9 cells were cultured in DMEM media containing 10% fetal bovine serum (FBS) and 1% penicillin–streptomycin in a tissue culture incubator at 37 °C with 5% CO$_2$. PCR mycoplasma detection kit (cat no. PM008, Shanghai Yise Medical Technology) was routinely used to test the mycoplasma contamination. The cells used in this study have given negative results in mycoplasma contamination test. SpCas9-expressing HEK293T (HEK293T-SpCas9) cells were generated by a PiggyBac transposon system followed by selection in the presence of 50 μg/ml hygromycin to ensure high Cas9 activity. HEK293T cells were transient transduced with pPB-TRE-spCas9-Hygromycin vector and pCMV-hybase vector with a 9:1 ratio to generate SpCas9-expressing HEK293T.

**Vector construction**. The LentiU6-LacZ-GFP-Puro (BB) vector was generated by our group previously (Addgene ID: 170459). This plasmid can also be acquired from the Luo lab (https://dream.au.dk/tools-and-resources).

**SURRO-seq library design**. Each SURRO-seq oligo consists of a BsmBI recognition site "cgtctc" with 4 bp specific nucleotides "acca" upstream, following the GGA cloning linker "aCACC", one bp "g" for initiating transcription from U6 promoter, 20 bp gRNA sequences of "gN20", 82 bp gRNA scaffold sequence, 37 bp surrogate target sequences (10 bp barcode sequences, 20 bp protospacer and 3 bp

PAM sequences, 4 bp downstream sequences), the downstream linker "GTTTg" and another BsmBI binding site and its downstream flanking sequences "acgg".

The SURRO-seq pool was designed as follows: (1) LibA contains 11 on target and corresponding 170 off target gRNAs from three published off-target detection methods (T7E1, GUIDE-seq, CIRCLE-seq); (2) LibB contains 110 gRNAs retrieved from published studies, which we expect to have sequence characteristics representative of gRNAs in gene therapy applications (cancers, PD-1, DMD, β-hemoglobinopathies, SCD, CCR5, HTT, CEP290). (3) We predicted off-target sites of each gRNA with FlashFry (v 1.80) and retrieved potential off-target with up to 4 bp mismatches in human genome hg19. (4) For each surrogate site, we added 10 bp barcode (fixed "AC" for the first two nucleotides + 8 bp Unique molecular identifiers (UMIs) sequences) to the upstream sequence of each selected gRNA, constructed the surrogate target sequence as 10 bp barcode + 23 bp gRNA (include PAM) + 4 bp downstream = 37 bp; (5) Off target sites with BsmBI recognition site were discarded, because of GGA cloning; (6) LibC contains surrogate sites with all possible 1 bp mismatch for five RGNs. The oligo pools were synthesized in Genscript® (Nanjing, China), and all sgRNA sequences and their oligos are provided in the Supplementary Data 2, 3, and 5.

**Construction of SURRO-seq plasmid library**. PCR amplification was used to amplify the 170-nt oligonucleotide pool. Firstly, the SURRO-seq oligos diluted to 1 ng/μl followed by PCR amplifications using the primers: SURRO (BsmBI GGA)-F and SURRO (BsmBI GGA)-R (Supplementary Data 6). The PCR reaction was carried out using PrimeSTAR HS DNA Polymerase (Takara, Japan) following the manufacturer's instruction.

The PCR products of SURRO-seq oligos were then used for Golden Gate Assembly (GGA) to generate the plasmids library. 36 parallel GGA reactions were performed, and the ligation products were pooled into one tube. Transformation was then carried out using chemically competent DH5α cells. For each reaction, 10 μl GGA ligation product was transformed in to 50 μl competent cells and all the transformed cells were plated on one LB plate (15 cm dish in diameter) with Xgal, IPTG and Amp selection. High ligation efficiency was determined by the presence of very few blue colonies. To ensure that there was sufficient coverage of each surrogate vector in the oligonucleotide library. For one library containing 12,000 synthetic oligos, 42 parallel transformations were performed, and all the bacterial colonies were scraped off and pooled together for plasmids midi-prep (PureLink™ HiPure Plasmid DNA Midiprep Kit, ThermoFisher Scientific). For small library, equal ratio reduction can be adjusted accordingly. For NGS-based quality quantification of library coverage, midi-prep plasmids were used as DNA templates for PCR amplifications, followed by gel purification and NGS sequencing.

**SURRO-seq plasmid library lentivirus packaging**. Supernatants containing lentiviral particles were produced by transient transfection of HEK293T cells using PEI 40,000 (Polyethylenimine Linear, MW 40,000). For 10 cm dish transfection, the DNA/PEI mixture contains 13 μg pLenti-SURRO-seq vectors, 3 μg pRSV-Rev, 3.75 μg pMD.2 G, 13 μg pMDGP-Lg/p-RRE, 100 μg PEI 40,000 solution (1 μg/μl in sterilized ddH$_2$O) and supplemented by Opti-MEM without phenol red (Invitrogen) to a final volume of 1 mL. The transfection mixture was pipetted up and down gently several times, and further incubated and kept at room temperature (RT) for 20 min. The transfection complex was added to 80%-confluent HEK293T cells in a 10-cm dish containing 10 ml of culture medium. After 48 h viral supernatant was harvested and filtered with a 0.45 μm filter. Polybrene solution (Sigma–Aldrich) was added to the crude virus solution to a final concentration of 8 μg/mL. The crude virus solution was aliquoted into 15 mL tubes (5 mL/tube) and store in −80 °C freezer until used.

**Lentivirus titer quantification by flow cytometry (FCM)**. The LentiU6-LacZ-GFP-Puro (BB) vector expresses an *EGFP* gene. The functional titer of our lentivirus prep was assayed by FCM (Supplementary Fig. S19). Briefly, (1) Day 1: Seed HEK293T cells to a 24-well plate. (2) Day 2: Transduce cells at 60~80% confluence. Before transduction, determine the total number of cells using one well of cells. The remaining wells were changed to fresh culture medium containing 8 μg/mL polybrene. A gradient volume of crude virus was added to each well and mix gently; (3) Day 3: Change to fresh medium without polybrene; (4) Day 4: Transduced cells were harvested with trypsin and washed with PBS twice. The suspended cells were fixed in 4% formalin solution at RT for 20 min. Cell pellet was washed with PBS twice and re-suspended in PBS solution, followed by FCM analysis. FCM was performed using a BD LSRFortessaTM cell analyzer with at least 30,000 events collected for each sample in duplicates. The FCM output data was analyzed by the software Flowjo vX.0.7. Percentage of GFP-positive cells was calculated as: Y% = N$_{GFP-positive cells}$ / N$_{total cells}$ × 100%. For accurate titter determination, there should be a linear relationship between the GFP positive percentages and crude volume. The titter (Transducing Units (TU/mL) calculation according to this formula: TU/mL = (N$_{initial}$ × Y% × 1000) / V. V represents the crude volume (μl) used for initial transduction.

**SURRO-seq library lentivirus transduction**. HEK293T-SpCas9 cells were cultured in D10 medium with 50 μg/ml hygromycin throughout the whole

experiment. For SURRO-seq library transduction, at Day −1: $2.5 \times 10^6$ cells per 10 cm dish were seeded. For a 12 K SURRO-seq library, transductions were performed in 10 replicates to reach 4000X coverage. For each group, one plate was used for cell number determination before transduction and another plate was used for drug-resistance (puromycin) test control. The remaining 10 plates were used for the SURRO-seq lentivirus library transduction (transduction coverage per gRNA exceeds 4000X of a 12 K library); 2) Day 0: We first determined the approximate cell number per dish. This was used to determine the volume of crude lentivirus used for transduction using a multiplicity of infection (MOI) of 0.3. The low MOI (0.3) ensured that most infected cells receive only 1 copy of the lentivirus construct with high probability. The calculation formula is $V = N \times 0.3 / TU$. $V$ = volume of crude lentivirus used for infection (ml); $N$ = cell number in the dish before infection; $TU$ = the titer of crude lentivirus (IFU/mL). The infected cells were cultured in a 37 °C incubator; 3) Day 1: 24 h after transduction, the cell was passaged at a ratio of 3 folds. 4) Day 2: The transduced cells were cultured in D10 medium containing 50 μg/ml hygromycin, 1 μg/mL puromycin, and 1 μg/mL doxycycline to induce Cas9 overexpression. 5) The transduced cells were spitted every 2~3 days when cell confluence reaches up to 90% at a ratio of 1:3. Cells from day 10 were harvested for further genomic DNA extraction. Parallel experiments were performed using wildtype HEK293T cells as MOCK controls.

**PCR amplicons of surrogate sites from cells**. Genomic DNA was extracted using the phenol-chloroform method. Then the genomic DNA was purified and subjected to SURRO-seq PCR. The PCR primers were SURRO-NGS-F and SURRO-NGS-R1 (Supplementary Data 6). In this study, 5 μg genomic DNA was used as template in one PCR reaction which contained approximately $7.6 \times 10^5$ copies of surrogate construct which covered about 63 times coverage of a 12 K SURRO-seq library. For each PCR reaction, briefly, 50 μl PCR reaction system consists of 5 μg genomic DNA, 0.5 μl PrimeSTAR polymerase (2.5 U/μl, R010A, Takara Bio), 4 μl dNTP (2.5 mM each), 10 μl PrimeSTAR buffer (5X, $Mg^{2+}$ Plus), 2.5 μl SURRO-NGS-F primer (10 μM), 2.5 μl SURRO-NGS-R1 primer (10 μM), and supplemented with ddH2O to a final volume of 50 μl. PCR reaction was carried out using a PCR thermal cycle using the following PCR program: 1 cycle at 94 °C for 2 min; 25 cycles of 94 °C for 20 s, 58 °C for 30 s, 68 °C for 45 s; and 1 cycle at 68 °C for 7 min. It is important that the PCR cycles were kept below 25 according to our optimizations. For a 12 K library, 32 parallel PCR reactions were performed to achieve approximately 2016 times coverage of each surrogate construct. Then the PCR products were purified by 1.5% gel and mixed with equal amounts and deep sequenced with a DNBseq sequencer.

**Synthetic gRNAs**. All synthetic gRNAs used for validation of OTs were chemically modified to increase stability in cells and synthesized by Synthego Co. (California).

**RNP nucleofection**. The CRISPR RNP was delivered into cells by nucleofection. For one nucleofection, 6 μg SpCas9 protein (Cat# 1081059, IDT) and the 3.2 μg synthetic gRNA (Synthego) was mixed in a PCR tube by pipetting and incubated at room temperature for at least 10 min and maximum 1 h. Then 200,000 suspended cells were gently resuspended cells in 20 uL nucleofection buffer (OptiMEM) by pipetting up and down. The cells and RNP complex were then transferred to a 4D-Nucleofector 16-well nucleocuvette strip (Catalog #: AXP-1004, Lonza). The samples should cover the bottom of the wells, and any presence of air bubble must be avoided. Nucleofection was performed with program CM-138. Immediately after electroporation, prewarmed culture media was added to the cells (180 μL per well of the Nucleocuvette strip). The cells were then transferred into one well of a 12-well cell culture plate with prewarmed medium. 48 h after transfection, cells were harvested for amplicon PCR and deep sequencing.

**Deep amplicon sequencing**. The on-target and off-target sites were amplified by PCRs. All primers used for PCR were showed in Supplementary Data 6. The amplicons were subjected to deep sequencing on the MGISEQ-2000 sequencer (MGI, China). All the samples were subjected to pair-ended 150 bp deep-sequencing on MGISEQ-2000 platform.

**Raw data processing**. FastaQC-v0.11.3 (https://www.bioinformatics.babraham.ac.uk/projects/fastqc/) and fastp-v0.19.6 (https://github.com/OpenGene/fastp) with default options were used for data quality control and filtering with the default parameters. The pair-end data was assembled using FLASH-v1.2.11 (http://www.cbcb.umd.edu/software/flash). BWA-MEM-v0.7.17 with default options was used to map the assembled data to the designed oligos sequence to preliminarily distinguish the data of each surrogate site.

**Data filtering**. The pysam module of Python-3.8 was used to split the aligned data according to the site number of the chip, and the reads of different sites were obtained. Then, we used three steps of strictly controlling parameters to filter the data of each site. Firstly, according to the structure of the chip, g + gRNA (20 bp) + scaffold (82 bp) + barcode (10 bp) + GTTT should remain unchanged at the beginning and end of each site. Then, to remove the chip synthesis errors, the pseudo editing sequences found in WT group were removed from spcas9 group.

Finally, to remove the interference of sequencing errors on the data, the extracted sequence of each site was re-aligned to the reference sequence, and the 1 bp indel on N1-N14 and N22-N27 of surrogate (27 bp) sequence were removed. The above three filtering steps were completed with julia-1.5.3 language.

**Fisher's exact test and statistical analysis**. To obtain stable and effective off-target efficiency, false positive results must be excluded. We used the number of reads of indel and no indel in spcas9 group and WT group to form a $2 \times 2$ matrix. Fisher's exact test was used to confirm whether the editing of each site was effective. To reduce False Discovery Rate (FDR), all $p$-values were corrected by BH (Benjamin and Hochberg) method. Next, we used strict parameters (Total read numbers(spCas9) ≥ 32, Indel read numbers (spCas9) ≥ 5, Indel Frequency (IF%) (WT) ≤ 25) to filter off-target efficiency with bias. Then we used parameters (Fold Change (FC) > 2, $p$-value (adjusted by BH) < 0.05) to divide the off-target data set into two parts for downstream analysis. The calculation formula of indel efficiency is as follows:

$$Indel\ Frequency(\%) = \frac{Indel\ read\ numbers}{Total\ read\ numbers} \times 100$$

And fold change is as follows:

$$FC = \frac{Indel\ efficiency\,[spCas9]}{Indel\ efficiency\,[WT]}$$

Fisher's exact test (two-sided, adjusted by BH) and other statistical analysis were performed in R-4.0.3. Visualization was completed by R and excel.

**Statistics & reproducibility**. In this study, Fisher's exact test (two-sided, adjusted by Benjamin and Hochberg) was used for testing the significance of RGN OT indels captured by SURRO-seq. One-way pair-wise ANOVA, unpaired T test (two-sided), paired T test (two-sided), Pearson and Spearman correlations (t-distribution testing for coefficient) were used for other statistical testing and indicated in figure legends. A $p$ value less than 0.05 is considered statistically significant. For all RGN OTs, we filtered OTs with low read coverage (total NGS reads less than 32). In SURRO-seq LibB, we filtered OTs potentially affecting cell growth based on enrichment: fold changes (SpCas9 NGS reads/MOCK NGS reads) > = 2 or depletion: fold changes (MOCK NGS reads/ SpCas9 NGS reads) > = 2. In this study, no statistical method was used to predetermine sample size. The experiments were not randomized. The Investigators were not blinded to allocation during experiments and outcome assessment.

**Reporting summary**. Further information on research design is available in the Nature Research Reporting Summary linked to this article.

## Data availability

All NGS data generated by this study have been shared via the CNGB public data depository with the following accession numbers: CNP0001979 and CNP0002648, and the Gene Expression Omnibus (GEO) data depository with the following accession number: GSE206347. A complete list of 702 NGS samples were summarized in Supplementary Data 7. All other relevant data supporting the key findings of this study are available within the article and its Supplementary Information files. Source data are provided with this paper.

## Code availability

The computing codes for analyzing indel frequencies with the deep sequencing from SURRO-seq has been deposited to GitHub[77]. URL to the codes: https://github.com/panxiaoguang/Massively_RGN_OTs/tree/v1.0.0.

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

## Acknowledgements

This project was partially supported by the Sanming Project of Medicine in Shenzhen (SZSM201612074, to L.B. and Y.L.), Qingdao-Europe Advanced Institute for Life Sciences Grant, Guangdong Science and Technology Department (No. 2019B1515120034), the Science, Technology, and Innovation Commission of Shenzhen Municipality (grant no. JCYJ20200109150410232), Guangdong Provincial Key Laboratory of Genome Read and Write (No. 2017B030301011 to X.Xu.), Guangdong Provincial Academician Workstation of BGI Synthetic Genomics (No. 2017B090904014 to X.Xu.), Danish Research Council (9041-00317B to J.G. and Y.L.), European Union's Horizon 2020 research and innovation program under grant agreement No 899417 (Y.L.), the Novo Nordisk Foundation (NNF21OC0068988 to J.G. and Y.L.; NNF21OC0071031 to Y.L.), the DFF Sapere Aude Starting grant (8048-00072 A to L.L.) and the National Human Genome Research Institute of the National Institutes of Health (RM1HG008525 to G.C.). We thank the China National GeneBank for the support of executing the project under the framework of Genome Read and Write.

## Author contributions

Conceptualization, Y.L.; Methodology, X.P., K.Q., X.X. and Y.L.; Investigation, X.P., K.Q., H.Y., X.X., C.A., L.P., X.L., P.H., G.I.C., F.X., P.L., J.Z., Y.Z., T.M. and Y.L.; Data Analysis, X.P., K.Q., H.Y., C.A., L.P., G.I.C. and Y.L.; Figure Preparation, X.P., K.Q., H.Y., X.X., C.A. and Y.L.; Writing – Original Draft, Y.L.; Writing – Review & Editing, all authors; Funding Acquisition, X.Xu., G.C., J.G., L.L. and Y.L.; Resources, H.J., J.L., J.W., N.J., L.B., H.Y., X.Xu., G.C., J.G., L.L. and Y.L.; Supervision, G.C., J.G., L.L. and Y.L.

## Competing interests
