## [Peer Review File · Nature Communications]

Reviewers' Comments:

Reviewer #1:

Remarks to the Author:

In this article, the authors describe a method called SURRO-seq, to use pooled lentiviral libraries to introduce many potential off-target sequences to cells (as surrogates of the endogenous genomic target sites) and enable testing in the presence of Cas9 to identify truly edited off-target sites. A concern is the novelty of the approach, since many reports (including some from these authors) have previously described lentiviral delivery of target sequences to measure editing potential of varied target sequences in cells. For example, the cited work of Fu et al Nat Communications 2022 (PMID: 35078987) is an extensive study using a dual target (on-target and off-target) design to explore off-target potential in a cellular context. Some analyses are performed which suggest that the energetics of the gRNA-DNA hybrid, mismatch type (especially G:U wobble) and mismatch position can all contribute to off-target potential; although again the novelty of these findings is somewhat limited by prior work, including from some of these authors. Nonetheless this method may offer a useful approach to evaluate a large number of off-target sites in a single experiment that can complement other off-target interrogation approaches.

1. The authors mention the lack of scalability of amplicon sequencing in the introduction (line #84) and throughout. They should cite and discuss rhAmpSeq as a much more scalable amplicon-seq method as compared to conventional amplicon sequencing (e.g. PMC7251314 and others).
2. The authors compare SURRO-seq to GUIDE-seq and CIRCLE-seq in Fig 2. I'm not convinced that off-targets found by one of GUIDE-seq and CIRCLE-seq but not by SURRO-seq can be considered false positives (Fig 2e). Alternatively these could be false negatives of SURRO-seq and one of the other methods. Amplicon-sequencing of the corresponding endogenous sites should be performed to resolve this.
3. Similarly what are the features of sites found by both GUIDE-seq and CIRCLE-seq but missed by SURRO-seq (like 4/55 for HEK293 site4 in Fig 2e)?
4. The methodologic description of the PCR step is limited. It mentions 5 ug genomic DNA is used but doesn't state volume of the PCR reaction. This amount of DNA seems to exceed usual guidelines for PCR. Are low-frequency alleles accurately detected in such a reaction with so much DNA?
5. Indel frequencies (Fig 3b, Fig S4a, etc.) should be shown on log₁₀ not linear scale, since indels in the low range (e.g. 0.1% to 1%) are of particular interest in the context of off-target editing.
6. What accounts for IF%, mock (Fig 2f,g)? This background would seem to limit the sensitivity of SURRO-seq to find low-frequency off-targets.
7. What is evidence that SURRO-seq can detect low indel frequency OTs that could not be captured by other methods (line 209)? Other methods like GUIDE-seq and CIRCLE-seq are typically highly sensitive and in fact in other places in manuscript it is claimed that these methods are overly sensitive (i.e. have false positives).
8. The selection criteria of the 23 OTs tested in Fig. 4 should be more clearly described. The data should be clearly shown comparing the indel frequency by SURRO-seq to the endogenous sequencing, both the data with lentiviral delivery of Cas9 (Fig 4a) and with RNP delivery of Cas9 (Fig 4e). I would suggest a scatter plot as a simple way to visualize endogenous vs SURRO-seq editing. If the SURRO-seq editing is present but the endogenous editing is absent, then it suggests that SURRO-seq may be identifying false positive sites with respect to gene editing the endogenous locus, which is not necessarily desirable for an off-target detection method.
9. How do the surrogate OTs impact cell proliferation by causing enrichment or depletion?
10. The authors should note that off-targets have previously been detected with 5 or 6 mismatches and with bulges between the DNA and RNA, so libraries evaluating all off-targets would need to be extremely large to be comprehensive (much larger than the up to 4 mismatch without bulges libraries tested). More practically existing methods such as in silico prediction and in vitro and/or cell-based empiric off-target detection assays would still need to be used before employing SURRO-seq to enable screening of moderate size libraries to define the off-target repertoire of a given RGN.

Reviewer #2:

Remarks to the Author:

Pan et al report a novel method for quantifying off-targets: SURRO-Seq. This method relies on providing defined surrogate Off-target sites (OTs) via a lentiviral vector and measuring indel formation using barcodes and deep sequencing. Importantly, the barcoding allows this method to overcome the indel split problem with OT sequencing. The authors compared the ability of this methodology to identify/quantify OTs identified previously from in vivo as well as in vitro approaches.

Overall the studies are well conceived and executed, and the manuscript is well written and easy to follow. The methodologies are rigorous and analyses sound. The method is validated in several ways including a comparison to nucleoprotein based editing.

The SURRO-Seq method will provide an alternative approach to measuring OT cutting frequencies in clinical RGNs. This work highlight the potential impact of OTs on oncogenes, which are important clinically.

A few minor comments

1. There are some places throughout the manuscript where the language and grammar should be reviewed.

Examples

"RGNs has vs RGNs have"

"to targeted evaluate"

"sentificity"

2. The authors have a very clear difference in on target editing and OT frequency as a function of cell type (4c/d) and while mentioned , this deserves a more thorough discussion. Particularly as contextual differences driving these results are unlikely due to gRNA sequence thermodynamics, which is he focus of Figure 5 and the discussion section.

3. Following up on point 2, while it is clear that the sequence/mismatch/thermodynamic models of the OT to gRNA is a contributor to the OT indel frequency (which has been previously reported) , it is also clear from these results (Fig 4 and 5) that other unexplained factors are equally important. One of the key values of this method is the ability to inform future studies to better elucidate factors affecting off target cleavage. I would challenge the authors to improve the discussion of how this approach may be used ot investigate or more fully elucidate factors outside of the gRNA sequence

REVIEWER COMMENTS

Reviewer #1 (Remarks to the Author):

In this article, the authors describe a method called SURRO-seq, to use pooled lentiviral libraries to introduce many potential off-target sequences to cells (as surrogates of the endogenous genomic target sites) and enable testing in the presence of Cas9 to identify truly edited off-target sites. A concern is the novelty of the approach, since many reports (including some from these authors) have previously described lentiviral delivery of target sequences to measure editing potential of varied target sequences in cells. For example, the cited work of Fu et al Nat Communications 2022 (PMID: 35078987) is an extensive study using a dual target (on-target and off-target) design to explore off-target potential in a cellular context. Some analyses are performed which suggest that the energetics of the gRNA-DNA hybrid, mismatch type (especially G:U wobble) and mismatch position can all contribute to off-target potential; although again the novelty of these findings is somewhat limited by prior work, including from some of these authors. Nonetheless, this method may offer a useful approach to evaluate a large number of off-target sites in a single experiment that can complement other off-target interrogation approaches.

Author response: We highly appreciate the reviewer's comments regarding the method, potential impact on the methodological novelty due to the already reported works by us and other groups in the field. From the first proof-of-concept point of view, we agree with the reviewer that using a pool of synthetic lentivirus library to capture the CRISPR-Cas editing footprints (indel profiles), on-target efficiency, and off-targets in cells have been demonstrated. The reported approach here, for simplicity we denoted it as SURRO-seq, is based on the all-in-one Golden Gate Assembly (GGA) vector (Addgene ID: 170459) generated by us previously. Our previous method was developed for capturing the CRISPR on-target efficiency. In this study, we introduce single molecular barcoding to enable the confident identification of indel reads from off-target sites different from 1-4 mismatches. This is a significant improvement of our previous method to broaden its applicability. The similar approach reported by Fu et al. 2022 is another excellent and similar strategy proving that synthetic surrogate sites can be used faithfully capture the on-target and off-target efficiency in cells. While the study is more focusing decoding the effect of sequence compositions and context on CRISPR-Cas9 activity, our study here used the similar approach to evaluate the off-target effects of therapeutic CRISPRs. Conceptually, Fu et al., and our approaches are comparable and collectively demonstrated that synthetic library-based approach is a very useful approach for targeted evaluating a large amount of potential off-target sites in cells in a single experiment. We appreciate the reviewer's agreement with the utility and value of such an approach for CRISPR off-target evaluations. In this revision, we have addressed this comment by including new analysis and/or highlight it in more thoroughly in the discussion.

1. The authors mention the lack of scalability of amplicon sequencing in the introduction (line #84) and throughout. They should cite and discuss rhAmpSeq as a much more scalable amplicon-seq method as compared to conventional amplicon sequencing (e.g.

PMC7251314 and others).

Author Response: Great thanks for the recommendation of highlighting rhAmpSeq library preparation technology which we missed to mention and discuss in our manuscript. Yes, we agree that rhAmpSeq is a very useful method for more scalable targeted amplicon sequencing. By designing and synthesizing hundreds of targeted PCR primers and performing two rounds of PCRs, the rhAmpSeq has showed great CRISPR off-target detection rate as compared to GUIDE-seq. In the revision, we have cited the rhAmpSeq, as well as including in **Table S1** and **Figure S1**.

An overview and comparison of methods for RGN off-target identification

Category 1	Category 2	Category 3
Genome-wide, cell-free screening: CIRCLE-seq Digenome-seq SITE-seq BLISS DIG-seq	Genome-wide, in-cell screening GUIDE-seq IDLV-seq HTGTS PEM-seq DISCOVER-seq	Targeted, in-cell validation rhAmpSeq* Deep-seq T7E1 TIDE ICE IDAA CUT-PCR

2. The authors compare SURRO-seq to GUIDE-seq and CIRCLE-seq in Fig 2. I'm not convinced that off-targets found by one of GUIDE-seq and CIRCLE-seq but not by SURRO-seq can be considered false positives (Fig 2e). Alternatively, these could be false negatives of SURRO-seq and one of the other methods. Amplicon-sequencing of the corresponding endogenous sites should be performed to resolve this.

RE: GUIDE-seq is based on the use of short double-strand oligonucleotide (dsODN) to label nuclease-induced DSBs. This has been one of the gold-standard genome-wide method for screening of CRISPR-induced off-targets in cells. In the original study by SQ Tsai et al., the authors have documented the potential detection of double-strand DNA breaks captured by GUIDE-seq, which is not dependent on the CRISPR off targets. Double-strand DNA breaks can occur during cell division, DNA replication errors in cells. Repair of these non-CRISPR-induced DSBs through the non-homologous end joining process (NHEJ) and the insertion of the targeted dsODN into the repaired DBS could lead to the detection of "off-target" site by GUIDE-seq. The CIRCLE-seq is a very conventional approach for genome-wide screening of CRISPR off-targets. The advantage of this method is that it is an in vitro cell-free assay method, which does not require the DNA repair machineries in cells. Daesik Kim and Jin-Soo Kim (Genome Research 2018. 28:1894-1900), who first reported the CIRCLE-seq method (cell-free, histone-free DNA), later in their DIG-seq (cell-free chromatin DNA) study found that chromatin can inhibit the Cas9 off-target effects. Only a small subset of off-target sites identified by CIRCLE-seq can be found by DIG-seq. We agree with the reviewer that since in our study, we did not perform further targeted amplicon-sequencing to address these non-overlap sites. The conclusion of false positive sites cannot be drawn. In this revision, we have

revised the conclusion for Fig. 2e, and provide a more thorough discussion regarding this issue. Revision in the main text is made in Page 5, line 140-150.

3. Similarly, what are the features of sites found by both GUIDE-seq and CIRCLE-seq but missed by SURRO-seq (like 4/55 for HEK293 site4 in Fig 2e)?

RE: Thanks for pointing out these four sites fail to captured by SURRO-seq. The features of these four non-detected off-target sites by SURRO-seq was shown in Supplementary Data 2. We also showed here for your reference. Based on the mismatch number and type of these four sites, we could not find a consistent pattern for the feature. Most likely, four sites are dropout detection sites by SURRO-seq, as the SURRO-seq method is based on. We included this potential dropout detection rate of SURRO-seq in the revised discussion.

4. The methodologic description of the PCR step is limited. It mentions 5µg genomic DNA is used but doesn't state volume of the PCR reaction. This amount of DNA seems to exceed usual guidelines for PCR. Are low-frequency alleles accurately detected in such a reaction with so much DNA?

Response: Thank you for pointing the PCR protocols. In the revision, we have included more detail regarding the PCR method. We indeed use 5µg genomic DNA in a 50 µl PCR reaction system. This method is optimized based on the previous CRISPR library based genetic perturbation method (Science. 2014 Jan 3; 343(6166): 84–87.), which used 10 µg genomics DNA in 100 µl PCR reaction system. This protocol has been optimized in our previous CRISPRon study (Nature Communications. 2021 May 28;12(1):3238.). The amount of input genomic DNA for each PCR reaction was calculated in order to increase the coverage of cells for the SURRO-seq library. For example, for a 12K SURRO-seq library, we performed 32 replicates of 50 ul reactions with 5 ug (equal to 200X coverage of a 12K SURRO-seq library) genomic DNA as template for each PCR reaction. It should be mentioned that, due to the large amount of genomic DNA included in each reaction, PCR conditions have to be optimized for the specific type of DNA polymerases. The SURRO-seq targeted PCR primers have been designed and optimized to maximize their specificity. For your reference, representative gel electrophoresis for the targeted PCR is shown below. The smear staining signal is from the genomic DNA template.

Representative gel electrophoresis picture:

- Indel frequencies (Fig 3b, Fig S4a, etc.) should be shown on log10 not linear scale, since indels in the low range (e.g. 0.1% to 1%) are of particular interest in the context of off-target editing.

Response: Thanks for the suggestion of change the scale of Figure 3b, S4a etc. into log10 form. For Fig 3b, the indel frequency of these on-target RGNs observed in MOCK cells are just background indel signals derived from either PCR errors and/or deep sequencing errors. Since this are not off-target editing, the presentation of the result for these on-target gRNA efficiencies is much nicer with linear scale (see the comparison below). For the indel frequency of RGN off-target sites (Fig. 4f and data 4.9), we have now used log2 scale for the plot.

Linear scale:

Log10 scale:

- What accounts for IF%, mock (Fig 2f,g)? This background would seem to limit the sensitivity of SURRO-seq to find low-frequency off-targets.

Response: The IF% found in mock samples is caused by background indels derived from the PCR and/or sequencing steps which are independent of CRISPR off-targets, also seen our response to point 5 above. This is indeed one limitation factor for the method and we have now highlighted it in the discussion section. The background indel frequency detected in MOCK lower the sensitivity of capturing off-target sites with low indel frequencies. We had already

used a low PCR cycle (25 cycles for each reaction) to reduce the PCR-introduced indels. The error rate of deep sequencing with DNBseq is about 0.1%.

7. What is evidence that SURRO-seq can detect low indel frequency OTs that could not be captured by other methods (line 209)? Other methods like GUIDE-seq and CIRCLE-seq are typically highly sensitive and in fact in other places in manuscript it is claimed that these methods are overly sensitive (i.e. have false positives).

Reply: Thanks for pointing out this overstatement/conclusion without systematically comparison. We have now revised our conclusion regarding the sensitivity as compared to other methods. As demonstrated with our data, the SURRO-seq offers a complementary, targeted, and high-throughput method for validation and evaluation of CRISPR off-targets in cells.

8. The selection criteria of the 23 OTs tested in Fig. 4 should be more clearly described. The data should be clearly shown comparing the indel frequency by SURRO-seq to the endogenous sequencing, both the data with lentiviral delivery of Cas9 (Fig 4a) and with RNP delivery of Cas9 (Fig 4e). I would suggest a **scatter plot** as a simple way to visualize endogenous vs SURRO-seq editing. If the SURRO-seq editing is present but the endogenous editing is absent, then it suggests that SURRO-seq may be identifying false positive sites with respect to gene editing the endogenous locus, which is not necessarily desirable for an off-target detection method.

Response: Thanks for this great suggestion of including scatter plot of the IF% measured by SURRO-seq vs. the IF% measured at the endogenous site by RNP delivery. We have also included the selection criteria for these 23 OTs selected for testing in the revised main text. Furthermore, Pearson correlation analysis for the IF% measured by SURRO-seq and the IF% measured at the corresponding endogenous sites showed good correlation, Pearson's R range from 0.88 to 0.94 in the five cell lines tested. We showed two plots Figure 4f and the full plots are provided in Supplementary data 4 (sub sheet 4.9).

f

9. How do the surrogate OTs impact cell proliferation by causing enrichment or depletion?

Response: In the SURRO-seq method, similar to other CRISPR pool gRNA libraries, the lentiviral vector expressing the individual CRISPR gRNAs are stably inserted into the targeted cells after drug (Puromycin for SURRO-seq) selection. The impacts on cell proliferation can arrive from several different mechanism. First, insertion mutagenesis caused the random integration of the lentiviral vector into essential genes (depletion) or cell fitness genes (enrichment). Second, gRNA with high off-targets can be depleted due to the simultaneous introduction of high dose of double-stranded DNA breaks. Third, OTs introduced in essential genes could affect cell proliferation as well. Reason for not including these enrichment or depletion sites is the concern that DNA sequence and mismatch features from these sites could affect the later investigation on how the DNA features influence RGN off-targets.

Taken these comments, we in the revision have analyzed and included the depletion/enrichment sites in supplementary figure S8 and Supplementary data 3 (Sheet Data 3.3).

For your reference, the updated Supplementary Figure S8 is shown below:

10. The authors should note that off-targets have previously been detected with 5 or 6 mismatches and with bulges between the DNA and RNA, so libraries evaluating all off-targets would need to be extremely large to be comprehensive (much larger than the up to 4 mismatches without bulges libraries tested). More practically existing methods such as in silico prediction and in vitro and/or cell-based empiric off-target detection assays would still need to be used before employing SURRO-seq to enable screening of moderate size libraries to define the off-target repertoire of a given RGN.

Reply: We agree that significantly detectable CRISPR-induced off-target indels have been observed and reported for some off-target sites with more than 4 mismatches and with bulges between the target site and the gRNAs. Our validation results (e.g., Figure 2G) also showed that some OTs with 5-6 mismatches still showed significantly detectable indels. And we agree with the reviewer that, the SURRO-seq provides a high throughput method for in-cell, targeted evaluation of CRISPR off-target cleavage. The selection of potential off-target sites should be based on in silico predictions, which we had already noted this point in the supplementary Note 1. Although the accuracy of currently available in silico prediction tools have been greatly improved, more RGN off-target data are still needed to further improve these RGN off-target predictors. And we are currently carrying out this investigation. We have included this valuable suggestion in the revised discussion.

Reviewer #2 (Remarks to the Author):

Pan et al report a novel method for quantifying off-targets: SURRO-Seq. This method relies on providing defined surrogate Off-target sites (OTs) via a lentiviral vector and measuring indel formation using barcodes and deep sequencing. Importantly, the barcoding allows this method to overcome the indel split problem with OT sequencing. The authors compared the ability of this methodology to identify/quantify OTs identified previously from in vivo as well as in vitro approaches.

Overall the studies are well conceived and executed, and the manuscript is well written and easy to follow. The methodologies are rigorous and analyses sound. The method is validated in several ways including a comparison to nucleoprotein based editing.

The SURRO-Seq method will provide an alternative approach to measuring OT cutting frequencies in clinical RGNs. This work highlight the potential impact of OTs on oncogenes, which are important clinically.

Response: We really appreciate the reviewer's time and effort in reviewing our manuscript and are glad to have the reviewer's strong recommendation and high evaluation of our method and its importance for the CRISPR gene editing and gene therapy field. Over the last 10 years, we have been working on developing and improving the CRISPR gene editing tools, as well as applying it therapy of human diseases and other biomedical applications, such as xenotransplantation. Currently, most genome-scale data generated are for CRISPR on-target efficiency. There is significantly lack of CRISPR off-target data, which greatly impact the

development better prediction tools for off-targets. We are glad to hear that the reviewer agrees with us that the SURRO-seq presented here provides an attractive, alternative and high-throughput strategy for evaluating CRISPR off-targets in cells. One important scalability of the number of gRNAs that we can evaluate simultaneously. Here we demonstrated the method for simultaneously evaluating over hundreds of RGNs and thousands of RGN OTs in one assay. We appreciate all the comments for the further improvement of our manuscript. In this revision, we have thoroughly addressed all these points.

A few minor comments

1. There are some places throughout the manuscript where the language and grammar should be reviewed.

Examples

“RGNs has vs RGNs have”

“to targeted evaluate”

“sentificity”

Response: Thanks for the correction. We have now asked a language editor to help with going through the language and grammar in the revised manuscript.

2. The authors have a very clear difference in on target editing and OT frequency as a function of cell type (4c/d) and while mentioned, this deserves a more thorough discussion. Particularly as contextual differences driving these results are unlikely due to gRNA sequence thermodynamics, which is the focus of Figure 5 and the discussion section.

Response: Thanks for point out this, which we had noticed as well. We agree that this cannot be explained by sequence context as they are the same when comparing between the five cell types tested. We had excluded the possibility of transfection (nucleofection) efficiency on the variabilities of on and off-target frequencies observed between different cell types. Possible explanations for this cell type-dependent variability can be caused by epigenetic factors (such as chromatin accessibilities at the target sites) and the different presence/preference of DNA repair machineries in these cell types. We had previously found that chromatin accessibility plays a significant role in affecting on-target frequency. Thank you again for highlighting this point. We have now included a section discussing the possible contribution of other sequence context-independent factors on the on- and off-target RNG activities.

3. Following up on point 2, while it is clear that the sequence/mismatch/thermodynamic models of the OT to gRNA is a contributor to the OT indel frequency (which has been previously reported), it is also clear from these results (Fig 4 and 5) that other unexplained factors are equally important. One of the key values of this method is the ability to inform future studies to better elucidate factors affecting off target cleavage. I would challenge the authors to improve the discussion of how this approach may be used to investigate or more fully elucidate factors outside of the gRNA sequence.

Response: Thanks for this great challenge. By addressing this challenge, it actually leads us to the formation of some ideas for further applying the SURRO-seq to elucidate factors outside the gRNA sequences on both on-target and off-target RGN activities. The revised discussion can be found in line 420-445. We also provide a graphical figure in Supplementary Figure S17-18.

Supplementary Figure S17. We propose a two-step cloning strategy to overcome the synthetic length limitation of the SURRO-seq oligonucleotide.

Supplementary Figure S17

Supplementary Figure S18. We provide some insightful suggestion on using the SURRO-seq to investigate the effects of epigenetic modification and DSB repair molecules on the on-target and off-target RGN activity.

Supplementary Figure S18

Reviewers' Comments:

Reviewer #1:

Remarks to the Author:

The authors have adequately responded to most of my specific comments in this revision. They acknowledge my overall concern about the novelty of the pooled lentiviral Cas9 target sequence approach.

One remaining point would be to mention in the discussion not only off-target sites with 5 or 6 mismatches but also off-target sites with DNA or RNA bulges are missing in the libraries tested (which tested ≤ 4 mismatches without bulges) and that these homologous bulge off-target sites represent potentially real sites that would not be captured by the current SURRO-seq implementation.

Reviewer #2:

Remarks to the Author:

The authors have adequately addressed my comments, as well as in my opinion the comments of the other reviewer in detail. They have done so either through additions/corrections to the main text or with added supplemental material. I particularly appreciate the added discussion on the value of this approach to study currently unknown factors impacting off-target activity.

REVIEWERS' COMMENTS

Reviewer #1 (Remarks to the Author):

The authors have adequately responded to most of my specific comments in this revision. They acknowledge my overall concern about the novelty of the pooled lentiviral Cas9 target sequence approach.

One remaining point would be to mention in the discussion not only off-target sites with 5 or 6 mismatches but also off-target sites with DNA or RNA bulges are missing in the libraries tested (which tested ≤ 4 mismatches without bulges) and that these homologous bulge off-target sites represent potentially real sites that would not be captured by the current SURRO-seq implementation.

Response: Thank you again for all the highly valuable comments and suggestions. Yes, in the current SURRO-seq implementation, we only select potential off-targets with up to 4 mismatches. Based on the previous genome-wide off-target screening results from e.g., GUIDE-seq and CIRCLE-seq, some off-target sites could have up to 6 mismatches and particularly off-target with DNA or RNA bulges. We have highlighted this limitation in the current library implementation.

Reviewer #2 (Remarks to the Author):

The authors have adequately addressed my comments, as well as in my opinion the comments of the other reviewer in detail. They have done so either through additions/corrections to the main text or with added supplemental material. I particularly appreciate the added discussion on the value of this approach to study currently unknown factors impacting off-target activity.

Response: We thank the reviewer again for all the valuable suggestions for improving the manuscript. We are also glad to hear that the reviewer appreciates the added discussion on how we can further apply the SURRO-seq to study the currently unknown factors affecting RGN off-targets.